# Impact of Formation Dip Angle and Wellbore Azimuth on Fracture Propagation for Shale Reservoir

**Kefeng Yang [1],\*, Lei Wang [2], Jingnan Ge [3], Jiayuan He [4], Ting Sun [5], Xinliang Wang [5] and Yanxin Zhao [2]**

1   China Petrochemical Corporation, Beijing 100728, China
2   School of Petroleum Engineering, Yangtze University, Wuhan 434000, China;
    wang-lei@yangtzeu.edu.cn (L.W.); wxl094315@outlook.com (Y.Z.)
3   PetroChina Zhejiang Oilfield Company, Hangzhou 311100, China; gejn85@petrochina.com.cn
4   Sinopec Petroleum Exploration and Production Development Research Institute, Beijing 102206, China;
    hejy.syky@sinopec.com
5   College of Safety and Ocean Engineering, China University of Petroleum, Beijing 100083, China;
    ting.sun@cup.edu.cn (T.S.); wxlcup@foxmail.com (X.W.)
\*   Correspondence: yangkefeng@sinopec.com

**Abstract:** The significant vertical heterogeneity, variations in ground stress directions, and irregular bedding interfaces make it extremely challenging to predict fracture propagation in continental shale reservoirs. In this article, we conducted a series of triaxial laboratory experiments on continental shale outcrop rocks to investigate the effects of formation dip angle and wellbore orientation on crack propagation under horizontal well conditions. Our study revealed that fracture propagation features can be categorized into four distinct types: (1) hydraulic fractures pass through the bedding interface without activating it; (2) fractures pass through and activate the bedding interface; (3) hydraulic fractures open and penetrate the bedding interface while also generating secondary fractures; and (4) hydraulic fractures open but do not penetrate the bedding interface. We found that as the dip angle decreases, the likelihood of fractures penetrating through the bedding interface increases. Conversely, as the dip angle increases, fractures are more likely to simply open the interface without penetrating it. Moreover, we observed that the well azimuth significantly affects the degree of fracture distortion. Specifically, higher azimuth angles corresponded to a higher degree of fracture distortion.

**Keywords:** continental shale; hydraulic fractures; formation dip angle; borehole azimuth

## 1. Introduction

China possesses abundant shale gas resources, estimated to be approximately 1.3 times greater than traditional gas reserves [1]. However, due to significant vertical heterogeneity, variations in formation stress directions, and irregular bedding interfaces, controlling fracture propagation in shale gas reservoirs proves to be extremely challenging [2]. Consequently, this results in low efficiency in enhancing the recovery factor. In order to gain a better understanding of the influence of formation dip angle and wellbore azimuth on fracture propagation, a series of true triaxial laboratory experiments were conducted in this study.

To date, numerous researchers have performed true triaxial experiments on outcrops to investigate the behavior of hydraulic fracture propagation. Some studies [3–10] utilized outcrops consisting of sandstone and sand-coal inter-beds. The findings indicate that when the fracture propagation reaches the interface, the fracture either ceases growth, changes its propagation direction, bifurcates into multiple directions, or continues to propagate along the original path and penetrates the bedding interface. The presence of weak cementation interfaces hinders the vertical propagation of hydraulic fractures, while high vertical stress differences and strong interfacial strength between sand layers facilitate vertical fracture

propagation. Additionally, when the horizontal stress difference reaches 3 MPa, hydraulic fracture propagation changes its direction, connecting to natural fractures in proximity to wells or weak cementation interfaces. Li Zhi et al. [11] conducted a study on the influence of bedding interfaces on fracture propagation. They highlighted the presence of wide openings and shear zones after the main fracture reached the bedding interface. These shear zones are typically longer than the opening zones and serve as the main channels for fluid flow. Altammar [12] and colleagues investigated the impact of vertical stress and interlayer properties on fracture height using cement samples measuring 30 cm × 30 cm × 10 cm. Their research demonstrated that the manner in which fractures penetrate through bedding interfaces depends on formation properties/conditions, and operational parameters. Sun Keming et al. [13,14] analyzed the influence of bedding dip angles and strength on fracture growth. Their findings indicated that when the main fractures, which experience the minimum vertical stress, reach the bedding interface, the smaller the angle between the bedding interface and fracture direction, the more likely the fracture is to change its original direction and extend along the bedding interface. Conversely, the larger the angle between the bedding interface and the original fracture direction, the greater the likelihood that the main fracture will penetrate the bedding interface and continue growing in its original direction. Zhou Tong [15], Liu Liming [16], Pang Tao [17], and others examined the impact of formation dip angles on fracture propagation through laboratory experiments and numerical simulations. They discovered that an increasing formation dip angle can restrict fracture growth in the vertical direction while enhancing the opening of the bedding interface. As the dip angle becomes larger, it becomes more challenging for the fracture to penetrate through the bedding interface. Jia Changgui, Hou Bing, and colleagues [18–22] conducted a series of true triaxial experiments on wells with high dip angles to study the effects of perforation phase angle, well dip angle, and well azimuth on initiating hydraulic fractures. Their results revealed that the well dip angle primarily influences the degree of fracture distortion, while the perforation phase angle mainly affects the number of hydraulic fractures. Furthermore, an increase in the azimuth angle leads to an increase in the degree of distortion for hydraulic fractures.

The existing literature primarily focuses on the impact of geological and engineering factors on hydraulic fracture propagation, particularly in relation to parallel bedding interfaces. However, there are few studies that have thoroughly examined the influence of bedding dip angles on fracture propagation. Furthermore, the majority of the existing literature concentrates on deviated wells rather than horizontal wells. As a result, this study aims to investigate the effects of formation dip angle and wellbore azimuth angle on fracture expansion behavior in continental shale. To accomplish this, we will conduct true triaxial physical simulation experiments on outcrop rock samples and subsequently clarify the fracture propagation law of continental shale. By evaluating fracture expansion behavior in relation to varying formation dip angles and wellbore azimuth angles, the study aims to provide comprehensive insights into the influence of these factors on fracture propagation. This research will contribute to a better understanding of hydraulic fracturing in continental shale formations and aid in optimizing drilling and fracturing operations.

## 2. Design of the Experiments

### 2.1. Experiment Protocol and Procedure

In this study, we utilized a large-scale true triaxial physical simulation system (refer to Figure 1) to investigate fracture propagation in continental shale outcrops. The selected outcrop samples were obtained from the Jiashengqi area in Anwen Town, Qijiang District, Chongqing City. These outcrop samples are representative of continental shale and exhibit favorable conditions, characterized by a high concentration of dark organic matter. To prepare the samples for experimentation, they were cut into 300 mm × 300 mm × 300 mm test specimens, as depicted in Figure 2. The preparation procedure for these samples is outlined in Figure 3. In the figure, $\alpha$ denotes the formation dip angle, which represents the

angle between the bedding interface and the horizontal plane. For a more comprehensive understanding of the parameters of the experimental rock samples, please refer to Figure 3.

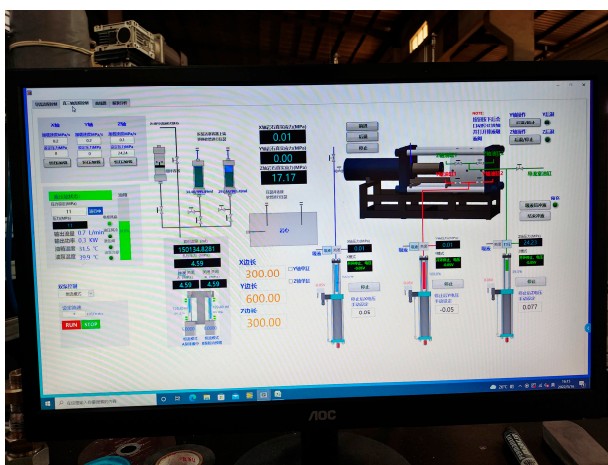

**Figure 1.** Large scale true triaxial physical simulation system.

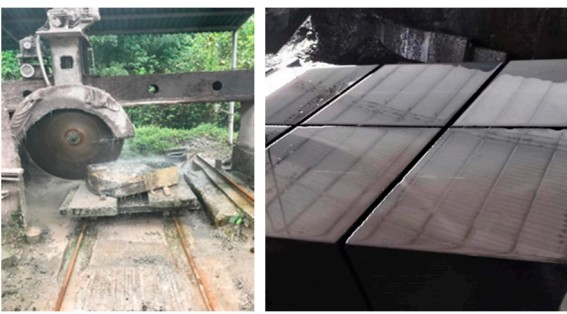

**Figure 2.** Process of cutting a shale outcrop.

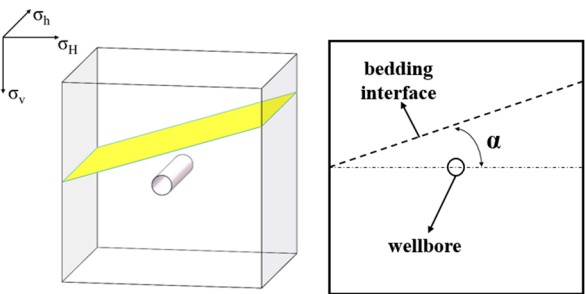

**Figure 3.** Schematic for preparing rock samples.

Figure 4 illustrates the four different drilling schemes employed in the study, each having distinct wellbore azimuths. Wellbore azimuth refers to the angle between the wellbore axis and the direction of the maximum horizontal principal stress. To simulate the actual drilling process, a drill bit with a diameter of 30 mm is used to bore the hole in accordance with the scheme depicted in Figure 4. Following the drilling process, the simulated wellbore is sealed using epoxy resin anchorage glue, as shown in Figure 5. In order to prevent the anchorage glue from obstructing the perforation holes, insulating tape is wrapped around the upper portion of the wellbore prior to sealing. The simulated wellbore possesses the following specifications: an outer diameter of 22 mm, an inner diameter of 10 mm, and a sealed bottom. The length of the wellbore can be adjusted based on the desired wellbore azimuth. Additionally, four 2 mm-diameter holes are drilled 10 mm above the bottom of the wellbore to simulate the presence of perforation holes.

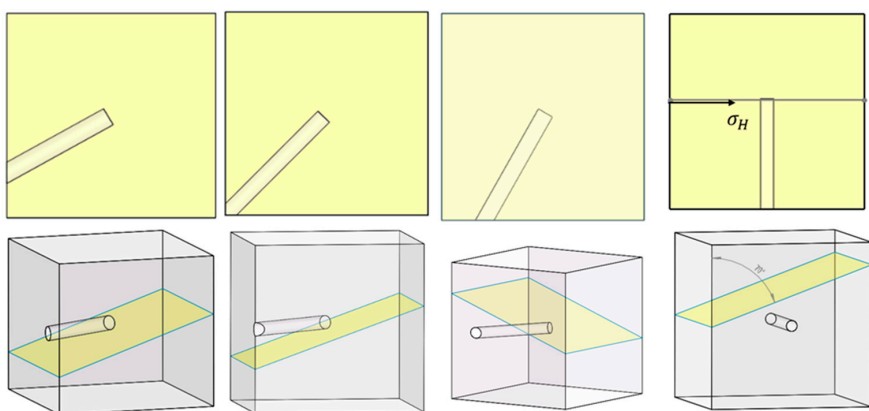

**Figure 4.** Drilling schemes for different wellbore azimuths.

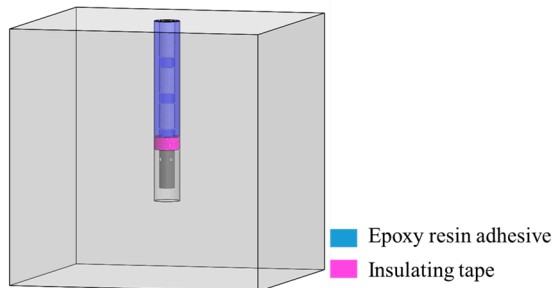

■ Epoxy resin adhesive
■ Insulating tape

**Figure 5.** Sealing and "well cementation" for rock samples.

## 2.2. Methodology of the Experiments

In this study, a total of nine experiments were conducted based on the similarity criterion to investigate the influence of the dip angle and wellbore azimuth angle on fracture propagation [23]. The objective of the first five sets of experiments was to explore the impact of different bed dip angles on fracture propagation. The remaining four sets of experiments specifically assessed the impact of altering the wellbore azimuth on fracture propagation. For more precise information on the experimental parameters, including the specific bed dip angles, wellbore azimuths, and their effects on fracture propagation, please refer to Table 1 in the paper.

**Table 1.** Experiment parameters.

| Rock Sample No. | Drainage m³/min | Viscosity mPa·s | $\sigma_h$ MPa | $\sigma_H$ MPa | $\sigma_v$ MPa | Formation Dip Angle | Wellbore Azimuth |
|---|---|---|---|---|---|---|---|
| Q1 | | | 17 | 30 | | 0 | |
| Q2 | | | 17 | 30 | | 10 | |
| Q3 | 0.0001 | 5 | 17 | 30 | 25 | 20 | 90 |
| Q4 | | | 17 | 30 | | 30 | |
| Q5 | | | 17 | 30 | | 40 | |
| F1 | | | 17 | 30 | | | 30 |
| F2 | | | 17 | 30 | | | 45 |
| F3 | 0.0001 | 5 | 17 | 30 | 25 | 20 | 60 |
| F4 | | | 17 | 30 | | | 90 |

## 2.3. Experimental Results

The experiment involved the use of fluorochrome to track the pattern of fracture propagation in samples. After the experiment, the samples were cut along the fracture propagation plane to observe the pattern, which was marked by fluorochrome. Table 2 and Figure 6 provide a summary of the laboratory results. It is evident from Figure 6 that the

main fracture in all samples initiated in a direction perpendicular to the wellbore. As it propagated vertically, it activated and opened at least two bedding interfaces and fully penetrated at least one bedding interface. In Figure 6, the left side represents the post-compression rock sample anatomy diagram, while the right side shows the recomposition of the fracture morphology. To clarify the color representation in the fracture morphology recomposition: the red color represents the main hydraulic fracture plane, which is the primary fracture that propagated through the rock sample. The blue color represents the activated interlayer interface, indicating the bedding interfaces that have been affected or opened by the hydraulic fracture. The green color represents the secondary hydraulic fracture plane generated by the activated interlayer interface. These secondary fractures occur along the activated bedding interfaces. The yellow color indicates the direction of the bedding interface within the rock sample. Based on Figure 6, the relationship between the main fracture and the bedding interface can be categorized into three distinct patterns: (1) Fracture Net with '十' Pattern: The hydraulic fracture opened and passed through the bedding interface, resulting in a fracture network resembling the shape of the Chinese character '十'. This pattern is depicted in Figure 6d,i.; (2) Fracture Net with 'T' Pattern: The hydraulic fracture opened the bedding interface but did not penetrate through it, forming a fracture network that resembles the shape of a letter 'T'. This pattern is illustrated in Figure 6g,h; and (3) Fracture Net with 'Y' Pattern: The hydraulic fracture opened and penetrated through the bedding interface, generating secondary hydraulic fractures along the bedding interface. This pattern forms a fracture network that resembles the shape of the letter 'Y'. Additionally, it should be noted that not all bedding interfaces in the rock samples were accurately identified. The observed bedding interfaces are those that were activated and opened during the experiment. It is speculated that there may be another pattern of fracture propagation net with an 'I' shape, where the hydraulic fracture passes through the bedding interface directly without activating it.

**Table 2.** Experiment results summary.

| Rock Sample No. | Summary of Fracture Propagation for All the Samples |
| :---: | :---: |
| Q1 | Having activated, opened, and penetrated 2 bedding interfaces and generated 1 secondary fracture. |
| Q2 | Having activated, opened, and penetrated 2 bedding interfaces and generated 2 secondary fractures. |
| Q3 | Having activated, opened, and penetrated 2 bedding interfaces and generated 1 secondary fracture. |
| Q4 | Having activated, opened, and penetrated 2 bedding interfaces |
| Q5 | Having activated, opened, and penetrated 2 bedding interfaces |
| F1 | Having activated, opened, and penetrated 2 bedding interfaces and generated 1 secondary fracture. |
| F2 | Having activated and opened, 2 bedding interfaces and penetrated only 1 bedding interface. |
| F3 | Having activated and opened, 2 bedding interfaces and penetrated only 1 bedding interface. |
| F4 | Having activated, opened, and penetrated 3 bedding interfaces. |

The use of fluorochrome in the experiment allowed for precise tracking and visualization of fracture propagation patterns. By cutting the samples along the fracture propagation plane and observing the marked patterns, researchers were able to gain deeper insights into the behavior of fractures in response to variations in bed dip angle and wellbore azimuth. Table 2 provides a comprehensive summary of the laboratory results, offering valuable information regarding the experiments conducted. The results presented in Figure 6 serve as graphical representations, displaying the anatomical and morphological characteristics of the fracture patterns. Analyzing Figure 6, it becomes evident that the main fracture consistently initiated perpendicular to the wellbore direction in all samples. The observation results show that the state of ground stress has a decisive influence on the initiation path and extension pattern of hydraulic fractures. As the fracture propagated vertically, it exhibited a remarkable ability to activate and open multiple bedding interfaces within the rock samples. It was also noted that the fracture fully penetrated at least one bedding interface, emphasizing the force and extent of its propagation. To aid in the interpretation of the fracture morphology recomposition in Figure 6, color-coded

visualizations were employed. The red color represents the main hydraulic fracture plane, which denotes the primary fracture path that propagated through the rock sample. The blue color represents the activated interlayer interface, signifying the bedding interfaces affected or opened by the hydraulic fracture. The green color represents the secondary hydraulic fracture plane generated along the activated bedding interface, indicating the occurrence of additional fractures parallel to the bedding interfaces. Finally, the yellow color represents the orientation of the bedding interface within the rock sample. Examining the relationship between the main fracture and the bedding interface, three distinct patterns emerged. The first pattern, known as the "Fracture Net with '十' Pattern," was characterized by the hydraulic fracture passing through the bedding interface, resulting in a fracture network resembling the shape of the Chinese character '十'. This pattern is exemplified in Figure 6d,i. The second pattern, referred to as the "Fracture Net with 'T' Pattern," was marked by the hydraulic fracture opening the bedding interface without complete penetration, forming a fracture network resembling the shape of the letter 'T'. This pattern is illustrated in Figure 6g,h. Finally, the third pattern, known as the "Fracture Net with 'Y' Pattern", featured the hydraulic fracture opening and penetrating the bedding interface, generating secondary hydraulic fractures along the bedding interface. This pattern forms a fractured network resembling the shape of the letter 'Y'. It is important to note that while the observed bedding interfaces were accurately identified in the rock samples, there may be additional patterns of fracture propagation that have not been thoroughly investigated. One potential pattern of interest is the "Fracture Net with 'I' Pattern", where the hydraulic fracture directly passes through the bedding interface without activating it.

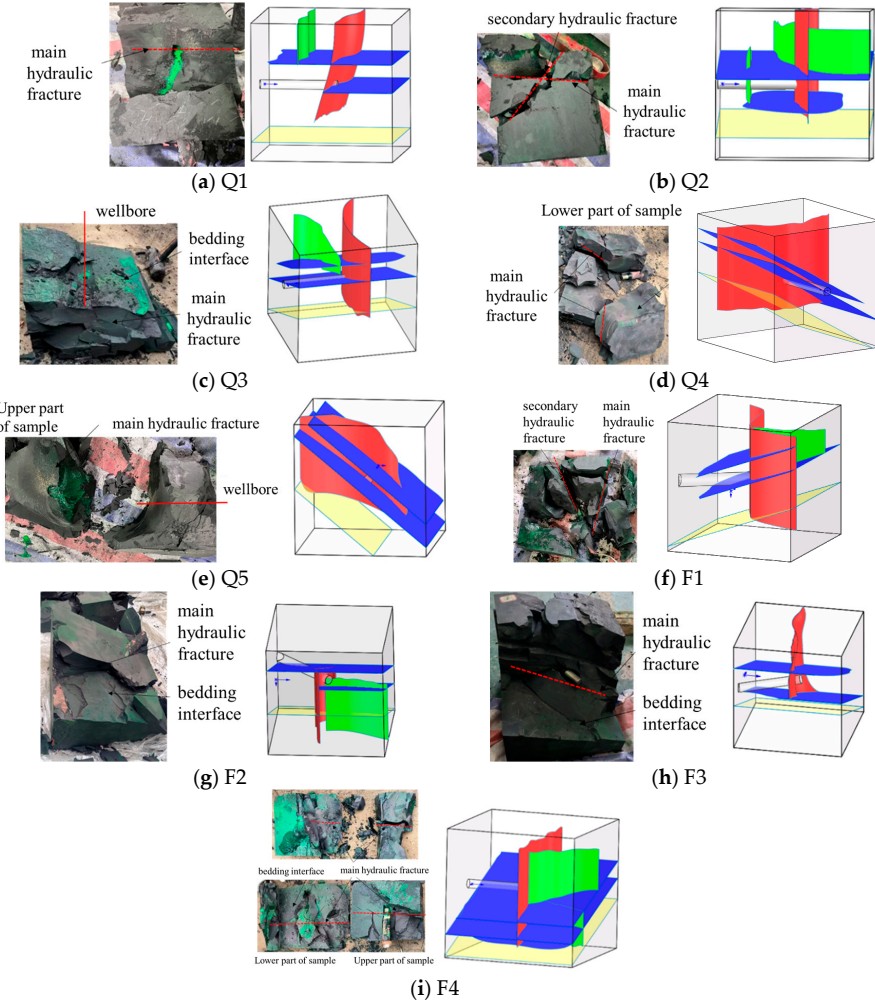

**Figure 6.** Experimental results under different formation dip angles.

Figure 7 provides a summary of the injection pressure curves observed in the experiments. Based on the characteristics of these pressure curves, we can identify at least two fracturing modes: (1) Peak Fracturing Pressure: Some injection pressure curves exhibit a distinct peak, indicating a clear fracturing event. This phenomenon is observed in rock samples Q4 and F3; and (2) No Peak Fracturing Pressure: On the other hand, other injection pressure curves do not show a clear peak, suggesting the presence of weak bedding interfaces around the perforated holes. In these cases, the hydraulic fractures initially propagate along these weak bedding interfaces, resulting in a lower fracturing pressure. This behavior can be observed in rock samples Q1 and F4. Additionally, by considering the extension pressure characteristics in combination with the observations of fracture propagation, we can identify two fracture extension modes: (1) Slowly Rising Trend: Some injection pressure curves exhibit a gradual and slow rise, indicating limited vertical fracture propagation. This trend is evident in samples Q3 and F1; and (2) Slowly Declining Trend: Conversely, other injection pressure curves demonstrate a gradual and slow decline, indicating unrestricted vertical fracture propagation. This trend can be observed in samples Q2 and Q5. Furthermore, sample Q3 exhibits a complex fracture pattern, which could be attributed to the presence of intercrossing fractures within the sample.

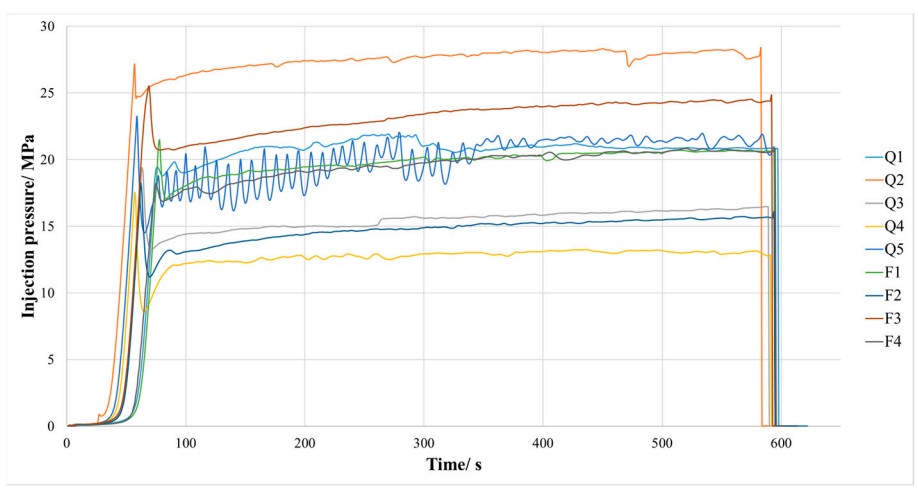

**Figure 7.** Comparison of the water-injection pressure curves.

## 3. Analysis of Impact Factors

### 3.1. Impact of Formation Dip Angles

The main objective of this section is to examine how changes in the dip angle of the formation impact the propagation of fractures. To achieve this, five experimental sets were conducted, with the formation dip angle being the only variable parameter while keeping all other factors constant. The formation dip angles tested were 0°, 10°, 20°, 30°, and 40°, as clearly stated in Table 1. The laboratory results obtained from the experiments are presented in Figure 6a–e, and a comprehensive comparison of these results can be found in Table 3.

**Table 3.** Comparison of the experimental results under different formation dip angles.

| Rock Sample No. | Depth of Fracture Extension/cm | Number of Activated Bedding Interfaces | Degree of Opened Bedding Interfaces | Number of Secondary Fractures |
|---|---|---|---|---|
| Q1 | 23.1 | 2 | 1.5 | 1 |
| Q2 | 22.6 | 2 | 1.5 | 2 |
| Q3 | 21.8 | 2 | 1.7 | 1 |
| Q4 | 20.5 | 2 | 1.7 | 1 |
| Q5 | 19.8 | 2 | 2 | / |

Based on the observations depicted in Figure 6 and summarized in Table 3, it can be inferred that the main fractures in all five samples were oriented perpendicular to the wellbore, with each fracture successfully penetrating two bedding interfaces. Analysis of the results revealed that as the formation dip angle increased, the opening of the bedding interfaces also increased, while the vertical length of the fractures decreased. Consequently, the study's conclusion states that small formation dip angles facilitate vertical extension of fractures and reduce the likelihood of activating the bedding interfaces. This phenomenon can be attributed to several factors. Firstly, when the vertical ground stress remains constant, higher dip angles result in lower stress on the bedding interfaces [24]. Consequently, the bedding interfaces become more susceptible to shear stress-induced destruction and are more easily activated through shear failure. This implies that fractures propagate along the weak bedding interfaces, reducing the vertical length of the fractures. Additionally, the higher the dip angle, the less likely it is for secondary fractures to occur on the activated bedding interface. This can be explained by the larger opening of the bedding interfaces, which leads to greater filtration loss and a decrease in water injection accumulation. A higher dip angle creates a larger opening, limiting the formation of secondary fractures and resulting in more focused fracture propagation.

In summary, when the formation dip angle is small (such as 0°), hydraulic fractures can smoothly extend vertically and penetrate the bedding interfaces with a lower likelihood of activation. Conversely, larger dip angles (greater than 40°) significantly increase the likelihood of activating the bedding interfaces, making it challenging for fractures to grow vertically. When the dip angle ranges from 10° to 30°, vertical fracture growth is comparatively easier, while the likelihood of activating the bedding interfaces is relatively higher, resulting in a more complex fracture propagation pattern.

The presence of natural fractures introduces various challenges for accurately predicting and controlling the formation and distribution of hydraulic fractures. These pre-existing fractures can create complex fracture networks as the injected fracturing fluid can propagate along these pathways, causing the hydraulic fractures to deviate from their intended trajectory. This can complicate the management of fracture geometry and the optimization of reservoir stimulation. Moreover, natural fractures contribute to fracturing fluid loss. Areas of poor fracture connectivity or high porosity can lead to fluid leakage, resulting in pressure loss and inefficient utilization of fracturing fluid. Apart from operational difficulties and increased costs, this fluid loss can also have adverse environmental impacts. However, it is important to note that natural fractures can also impede the vertical expansion of hydraulic fractures, limiting their ability to adequately communicate with high-quality reservoirs. If a natural fracture intersects vertically with a hydraulic fracture or is shielded within the target zone, the hydraulic fracture may not be able to extend effectively into the reservoir, reducing the overall efficiency and productivity of the fracturing operation.

Therefore, it becomes crucial to thoroughly evaluate the geological characteristics of the formation, understand the nature of the fracture network, and consider the operational requirements when designing fracturing strategies. By analyzing field geological data and incorporating the geological model, it becomes possible to optimize the vertical expansion of hydraulic fractures, improve the connectivity of natural fractures, and shape a complex fracture network. This optimization can be achieved through careful adjustment of injection pressures, construction parameters, and the sequence of fracturing fluid injection.

### 3.2. Impact of Wellbore Azimuth Angles

In order to thoroughly investigate the influence of wellbore azimuth on fracture propagation, a series of meticulously designed laboratory experiments were conducted. These experiments aimed to vary the wellbore azimuth while maintaining all other parameters constant, thereby isolating the effect of azimuth on fracture behavior. The tested wellbore azimuths were 30°, 45°, 60°, and 90°, as specified. The results obtained from these exper-

iments are illustrated in Figure 6f–i, and a comparison of the experimental outcomes can be found in Table 4.

**Table 4.** Comparison of the experimental results under different wellbore azimuth angles.

| Rock Sample No. | Amplitude/MPa | Distortion Length/cm | Depth of Fracture Extension/cm | The Number of Penetrated Layers/Layer |
|---|---|---|---|---|
| F1 | 4.43 | 20.6 | 21.5 | 2 |
| F2 | 3.56 | 9.6 | 17.8 | 1 |
| F3 | 3.65 | 8.2 | 20.3 | 1 |
| F4 | 3.93 | 0 | 20.8 | 3 |

Based on the observations depicted in Figure 6 and summarized in Table 4, it can be concluded that the fractures consistently initiated perpendicular to the wellbore, regardless of the tested wellbore azimuths. However, variations in wellbore azimuth did exhibit significant differences in fracture behavior, specifically in terms of distortion level and initiation direction. The length of the fractures did not show a significant variation with different wellbore azimuths. However, the distortion level of the fractures was found to be notably different. When the wellbore azimuth was smaller, the distortion level of the fracture increased. This can be attributed to the fracture initially propagating perpendicular to the wellbore. However, due to existing ground stress, the fracture changed its propagation direction to align along the maximum horizontal formation stress. As a result, the fracture experienced higher distortion when the wellbore azimuth was smaller. In contrast, when the wellbore azimuth was higher, fracture initiation occurred along the maximum horizontal formation stress, and the fracture continued to propagate in that direction without changing its trajectory. This resulted in less distortion of the fracture compared to smaller wellbore azimuths. Furthermore, it is worth noting that smaller wellbore azimuths required higher formation fracture pressures. This is because when the fracture initiates perpendicular to the wellbore, it needs to overcome higher stress to propagate and crack the formation.

In summary, the wellbore azimuth has a significant impact on the distortion level and initiation direction of fractures. Smaller wellbore azimuths result in higher distortion levels as the fracture changes its trajectory, while larger wellbore azimuths lead to fractures initiating and propagating parallel to the maximum horizontal formation stress. Additionally, smaller wellbore azimuths require higher formation fracture pressures due to the increased stress the fracture needs to overcome to propagate.

Therefore, it is crucial to integrate geological data in order to optimize the location of drilling before commencing the drilling process. By taking into account various factors such as geological characteristics, lithology, fault distribution, and other relevant data, it becomes possible to identify the optimal position for the borehole, ensuring a matching trajectory with the stress distribution within the formation. Through judicious selection of the relationship between the borehole trajectory and ground stress, fluid flow can be enhanced, the challenges associated with fracturing construction can be reduced, and the effectiveness of hydraulic fractures can be improved.

## 4. Conclusions

The paper utilized outcrop rock samples to conduct laboratory experiments, focusing on the influence of formation dip angle and wellbore azimuth on fracture propagation under horizontal well conditions. The key conclusions drawn from the study are as follows:

(1) Natural Fractures: The presence of natural fractures poses challenges in predicting and controlling hydraulic fracturing. It can result in multiple propagation characteristics of hydraulic fractures, which can be categorized into four types. These include fractures passing through bedding interfaces without activating them, fractures acti-

vating and opening bedding interfaces, fractures opening and passing through bedding interfaces while generating secondary fractures within them, and fractures opening bedding interfaces without penetrating them;

(2) Formation Dip Angle: A smaller dip angle of the formation leads to a greater vertical extension of the hydraulic fracture, making it easier to penetrate through interfaces. Additionally, a smaller dip angle reduces the likelihood of activating and opening bedding interfaces. On the other hand, a larger wellbore azimuth results in smoother fracture patterns and easier vertical propagation. However, when the formation dip angle falls within the range of 10–30°, the fracture behavior becomes more complex due to the balanced probability of the hydraulic fracture both opening and penetrating the bedding interfaces;

(3) Design Considerations: Prior to drilling, considering factors such as geological characteristics, lithology, and fault distribution can help determine the optimal drilling location. This aids in reducing the difficulties encountered during hydraulic fracturing operations. When designing hydraulic fracturing construction plans, it is essential to optimize construction parameters and pumping schedules based on a comprehensive evaluation of various factors. This approach facilitates achieving the optimal expansion of hydraulic fractures and shaping the fracture networks.

**Author Contributions:** Conceptualization, K.Y.; Methodology, K.Y. and L.W.; Investigation, K.Y. and L.W.; Resources, K.Y.; Writing—original draft, K.Y., L.W., J.H. and T.S.; Writing—review & editing, K.Y., L.W., J.G., J.H., T.S., X.W. and Y.Z. All authors have read and agreed to the published version of the manuscript.

**Funding:** This research was funded by Research on drilling and production enhancement technology of deep coalbed methane (Grant number P23207).

**Data Availability Statement:** No new data were created.

**Conflicts of Interest:** The authors declare no conflict of interest.

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
