# Peer review of "Impact of Formation Dip Angle and Wellbore Azimuth on Fracture Propagation for Shale Reservoir"

_processes, doi:10.3390/pr11082419_

Round 1

Reviewer 1 Report

Thank you very much for your kind invitation.

The interplay between structural surfaces and fluids has been extensively discussed. In particular, revealing the controls on the propagation of hydraulic fractures in shale strata is a centric topic for both petroleum and structural geological societies. This manuscript presents a lab-obtained dataset that concerns mainly the impact of formation dip angle and wellbore azimuth on fracture propagation. The authors collected terrestrial shale samples from Anwen Town, Qijiang District, and Chongqing City. They placed the planar samples in different dip angles in a large-scale tri-axial physical simulation system and drilled in at different azimuths to induce fracture propagation. Then, they observed the propagation patterns and sorted them into four categories.

Generally, the experiments are well-designed and the results provide valuable data for future studies. The conclusions are generally sound and the manuscript is easy to follow. I recommend that the paper could be accepted after some minor revisions.

Comments:

1. The authors should describe the samples in detail, including their locations, lithologies, grain size, and the nature of cement.

2. The figure labels should be translated into English.

3. The references should be checked and reformatted. For instance, references Nos. 9 and 16 are capitalized, while the others are not.

Author Response

  1. The authors should describe the samples in detail, including their locations, lithologies, grain size, and the nature of cement.

Reply: A brief description of outcrop rock samples has been added to the article.

  1. The figure labels should be translated into English.

Reply: The labels in the image have been corrected to English format.

  1. The references should be checked and reformatted. For instance, references Nos. 9 and 16 are capitalized, while the others are not.

Reply: The format of the references has been standardized.

Reviewer 2 Report

This paper conducted tri-axial laboratory experiments on continental shale outcrop rocks with various formation dip angles and wellbore azimuths to understand fracture propagation in such reservoirs. The research identified four fracture propagation categories and found that a smaller dip angle increases the likelihood of fractures penetrating the bedding interface, while a higher dip angle makes it more probable for fractures to open the interface. The paper topic is very interesting and I find it a very good match with the scope of the journal. However, I have some minor comments to improve the quality of the paper.

1. One critical aspect that requires attention is the need for a more comprehensive elaboration on the novelty of your paper. In the abstract, it would be beneficial to provide a detailed account of the unique contributions your research brings to the field, setting it apart from similar studies. This will assist readers in understanding the distinctive value your work offers within the context of existing research.

2. Consider enhancing the quality of Figures 1 and 2. Figure 1 appears to be a screenshot captured from a computer screen, and it would greatly benefit from improvement in its visual clarity.

3. Please ensure that all the text in the paper is presented in English. For instance, Figures 3, 5, 6, and 7 requires modification.

4. enhance the Introduction section by providing a more extensive discussion on prior studies conducted in the field. It is recommended to elevate the quality and comprehensiveness of the introduction by referencing the following paper: https://doi.org/10.1016/j.ijggc.2023.103920 , https://doi.org/10.1080/10916466.2020.1780256 

5. provide additional elaboration on sealing and well cementation concerning the rock samples? 

6. Kindly consider enhancing the linguistic quality of the paper as it could significantly augment its readability. The presence of specific editorial errors, such as those identified in Section 2.3, should be addressed to refine the overall presentation further.

7. I would recommend that the conclusion begin by elucidating the novelty of the study, followed by a comprehensive discussion on the influence of formation dip angle and wellbore azimuth angle.

8. In Table 2, further elaboration on the fracture propagation for all the samples would be beneficial. Additionally, considering the utilization of figures with a DPI of 300 and above, particularly for Figure 6, is recommended.

The paper is highly engaging and aligned with the scope of the journal. Nevertheless, I firmly believe that implementing the aforementioned comments can substantially enhance the paper's quality.

Author Response

  1. One critical aspect that requires attention is the need for a more comprehensive elaboration on the novelty of your paper. In the abstract, it would be beneficial to provide a detailed account of the unique contributions your research brings to the field, setting it apart from similar studies. This will assist readers in understanding the distinctive value your work offers within the context of existing research.

Reply: The novelty description has been added to the abstract.

  1. Consider enhancing the quality of Figures 1 and 2. Figure 1 appears to be a screenshot captured from a computer screen, and it would greatly benefit from improvement in its visual clarity.

Reply: Changes have been made to the clarity of the image.

  1. Please ensure that all the text in the paper is presented in English. For instance, Figures 3, 5, 6, and 7 requires modification.

Reply: The labels in the image have been corrected to English format.

  1. enhance the Introduction section by providing a more extensive discussion on prior studies conducted in the field. It is recommended to elevate the quality and comprehensiveness of the introduction by referencing the following paper: https://doi.org/10.1016/j.ijggc.2023.103920 , https://doi.org/10.1080/10916466.2020.1780256

Reply: The citation of relevant literature has been added.

  1. provide additional elaboration on sealing and well cementation concerning the rock samples?

Reply: This article uses epoxy resin adhesive to seal and fix the wellbore, and applies electrical tape to ensure that the perforated wellbore is not blocked.

  1. Kindly consider enhancing the linguistic quality of the paper as it could significantly augment its readability. The presence of specific editorial errors, such as those identified in Section 2.3, should be addressed to refine the overall presentation further.

Reply: The language in the article has been polished and modified.

  1. I would recommend that the conclusion begin by elucidating the novelty of the study, followed by a comprehensive discussion on the influence of formation dip angle and wellbore azimuth angle.

Reply: The conclusion has been modified.

  1. In Table 2, further elaboration on the fracture propagation for all the samples would be beneficial. Additionally, considering the utilization of figures with a DPI of 300 and above, particularly for Figure 6, is recommended.The paper is highly engaging and aligned with the scope of the journal. Nevertheless, I firmly believe that implementing the aforementioned comments can substantially enhance the paper's quality.

Reply: The image has been modified.